# Prevalence, Predictors, and Experience of Moral Suffering in Nursing and Care Home Staff during the COVID-19 Pandemic: A Mixed-Methods Systematic Review

**DOI:** 10.3390/ijerph19159593

**Published:** 2022-08-04

**Authors:** Zainab Laher, Noelle Robertson, Fawn Harrad-Hyde, Ceri R. Jones

**Affiliations:** 1Department of Neuroscience, Psychology and Behaviour, University of Leicester, Leicester LE1 7HA, UK; 2Department of Health Sciences, University of Leicester, Leicester LE1 7RH, UK

**Keywords:** moral injury, moral distress, mental health, workplace well-being, healthcare workers, nursing home staff, care home staff, COVID-19

## Abstract

(1) Background: Nursing and care home staff experienced high death rates of older residents and increased occupational and psychosocial pressures during the COVID-19 pandemic. The literature has previously found this group to be at risk of developing mental health conditions, moral injury (MI), and moral distress (MD). The latter two terms refer to the perceived ethical wrongdoing which contravenes an individual’s moral beliefs and elicits adverse emotional responses. (2) Method: A systematic review was conducted to explore the prevalence, predictors, and psychological experience of MI and MD in the aforementioned population during the COVID-19 pandemic. The databases CINAHL, APA PsychINFO, APA PsychArticles, Web of Science, Medline, and Scopus were systematically searched for original research studies of all designs, published in English, with no geographical restrictions, and dating from when COVID-19 was declared a public health emergency on the 30 January 2020 to the 3 January 2022. Out of 531 studies screened for eligibility, 8 studies were selected for review. A thematic analysis was undertaken to examine the major underpinning themes. (3) Results: MI, MD, and related constructs (notably secondary traumatic stress) were evidenced to be present in staff, although most studies did not explore the prevalence or predictors. The elicited major themes were resource deficits, role challenges, communication and leadership, and emotional and psychosocial consequences. (4) Conclusions: Our findings suggest that moral injury and moral distress were likely to be present prior to COVID-19 but have been exacerbated by the pandemic. Whilst studies were generally of high quality, the dearth of quantitative studies assessing prevalence and predictors suggests a research need, enabling the exploration of causal relationships between variables. However, the implied presence of MI and MD warrants intervention developments and workplace support for nursing and care home staff.

## 1. Introduction

The World Health Organization (WHO) declared COVID-19 a public health emergency of international concern (PHEIC) on 30 January 2020 and categorized it by 11 March 2020 as a global pandemic. To date, the WHO has reported over a quarter of a billion known cases and over five million cumulative deaths worldwide [1]. The respiratory disease, caused by severe acute respiratory syndrome coronavirus 2 (SARS-CoV-2), has had a profound global impact on individuals and systems, changing the landscape of everything in its path [2].

Nursing and care homes (N&CH) have been particularly affected by the virus [3]. N&CH, in the context of this review, refers to establishments providing long-term, 24-h residential care to older adults who are passed retirement age (60 years old and over), and includes nursing homes that provide onsite medical care support in addition to social and personal care, and other residential care homes which do not provide onsite medical care [4]. N&CH provision and organizational structure may also vary nationally and internationally.

Frailties associated with advanced age and pre-existing comorbidities such as cardiovascular disease, diabetes, and dementia put older adults at high risk of being infected by COVID-19 and, in turn, experiencing significantly higher morbidity and mortality rates than other demographics [2,5]. These vulnerabilities have meant that N&CH employees have been exposed to unparalleled challenges during the pandemic, in addition to “the perfect storm” of antecedent pressures such as strained resources [3,6]. The Office for National Statistics (ONS) reported, in 2021, an increase of 19.5% deaths in the UK’s care sector alone during the pandemic compared with the five-year average [7]. The ONS recorded 40,962 N&CH resident deaths in England and Wales wherein COVID-19 was the leading cause of death and a total of 42,189 N&CH resident deaths throughout all waves of the pandemic wherein COVID-19 was either confirmed or suspected to be involved [7].

Subsequently, research has begun to explore the psychological and philosophical implications of COVID-19 in the health and social care workforce (e.g., N&CH staff) through the concept of moral injury (MI) and moral distress (MD) [8,9].

MI can be defined as occurring after events and situations that are perpetrated, experienced, witnessed, learned about, or not prevented, involving perceived betrayal, a sense of injustice, and/or non-support (particularly from those who hold legitimate power, e.g., leaders); this experience may then compromise an individual’s deeply held beliefs and values [10,11]. The act of omission (inaction) or commission (action) during and following potentially morally injurious events (PMIEs) may be perpetrated by oneself or by others [12,13], and it may be that an individual does not need to cognitively acknowledge or understand what a “right” or “ethical” decision or action would be for them to experience MI or MD [14]. Guilt, anger, and anxiety appear as common emotional corollaries of MI and MD which may then contribute to or exacerbate mental health difficulties [9,10].

As an asserted precursor to MI, MD denotes the psychological pain an individual feels following inaction or incorrect action, through perceived internal or external factors, in a situation that has ethical and moral dimensions [15,16,17]. Both MI and MD are often used interchangeably in the literature, often have attributable similarity in definitions, and have slight subtleties in their differences, as well as limited scientific exploration of construct overlap and/or difference [12,18]. It has been posited that MI and MD significantly differ in the nature of their emotional consequences, duration, and outcome [12]. Whilst causes for both usually relate to events with ethical and moral implications, MD is considered to be a transient emotional disturbance that does not cause lasting emotional harm, whereas MI is described as a deeper “emotional wound” of longer duration, and, if left unmanaged, it can potentiate or aggravate mental health difficulties [12,13,18].

In the context of COVID-19, some sources of PMIEs may be the inadequate resources to manage the pandemic, thus putting people at risk; the experience of frequent fatalities; the status degradation and perceived unjust treatment from others (e.g., superiors); and the shirking of responsibility and accountability [13,19].

The exploration of MI and MD in healthcare professionals (HCPs) in medical hospital settings during COVID-19 predominates in the published literature, revealing an elevation in both constructs. A recent mixed-methods cross-sectional study found that 41% of HCPs reporting scores on the Moral Injury Symptom Scale Health Professional (MISS-HP) were clinically significant, meaning that they had a MISS-HP score of 36 and above, which is determined as the clinical cutoff signifying the presence of MI [20]. Similarly, a Romanian study using an adapted version of the Moral Injury Events Scale (MIES), where higher scores are thought to indicate increased intensity and prevalence of PMIEs, reported that 46.8% of physicians working in medical units scored above the median MIES score, scoring above 24 out of a possible score total of 45 [21]. They also found a significant association between PMIEs experienced and physical and psychological impact [21]. Qualitative and mixed-methods research studies exploring MI in hospital staff have reported common major themes of organizational and infrastructural issues; advocating difficulties for the care of patients; and care burden, where HCPs expressed their distrust of leadership and feelings of non-support [20,22].

Whilst the literature of MI and MD in hospital-based HCPs is growing, the number of MI studies conducted in N&CHs, often labelled the “forgotten sector”, remains limited [23]. In a mixed-methods study conducted before the pandemic, researchers found a significant association between frequent exposure of PMIEs and MD severity in N&CH staff caring for older adults with dementia [24]. In the same study, semi-structured interview data revealed that insufficient staffing levels was most frequently identified as the source of PMIE. Other qualitative studies conducted in N&CHs predating COVID-19 have reported themes of conflicting ethical principles and expectations regarding care, lack of resources, struggles with staff autonomy, perceived powerlessness of staff against leadership, and communication difficulties as sources of MI and MD [23,25,26,27].

The identified gap in empirical studies may be attributed to the lack of consensus in defining what MI and MD are, how they are distinguishable from other mental health difficulties, and the limited evidence on the relationship between these and related concepts [12].

Despite N&CH staff being under immense and unremitting pressures, the subject of MI and MD in the N&CH population during COVID-19 is relatively unexplored [3,28] in comparison to forms of anxiety; the prevalence of moderate-to-severe posttraumatic stress disorder (PTSD) and/or anxiety symptomology in N&CH staff during the first wave of the pandemic has been estimated to be as high as 43% [29]. Due to the implications this has on N&CH staff mental health and well-being, and in turn service delivery and quality of care provided to residents, the identified research space necessitates further exploration [9,30]. This systematic review explores the prevalence, experience of, and psychological impacts of MI and MD in N&CH staff during the COVID-19 pandemic.

The primary aim of this review is to evaluate the existing literature of MI and MD in N&CH staff to address the following questions:What is the estimated prevalence of MI and MD?What factors increase the likelihood of developing MI and MD?What is the psychological impact and general experience of MI and MD?

## 2. Materials and Methods

### 2.1. Main Method

Our review methodology was guided by Bettany-Saltikov [31] and the Cochrane Handbook [32] and reported in line with PRISMA guidelines [33]. A mixed-method approach was adopted, whereby original quantitative, qualitative, and mixed design studies were reviewed, analyzed, and integrated together to inform a comprehensive understanding of MI and MD in N&CH staff in accordance with Reference [32]. This approach allowed us to address the review aims by consolidating complementary information [32]. PROSPERO was consulted to check for preregistered reviews in this area to ensure there were no significant overlaps with any anticipated reviews. The current review was not preregistered, due to COVID-19-related delays in the preregistration process and the speed in which our review needed to be generated. The review process consisted of an initial scoping search of MI and MD, defining the research question and eligibility criteria, study selection, appraisal and extraction, and data analysis and synthesis [31]. The review process was conducted independently by one reviewer (ZL) and supervised by experienced psychologists at the University of Leicester’s Department of Neuroscience, Psychology and Behavior (NR, FHH, and CRJ).

### 2.2. Information Sources and Search Strategy

Electronic databases were used as information sources for our review. The indexes Cumulative Index to Nursing and Allied Health Literature (CINAHL), APA PsychINFO and APA PsychArticles, Web of Science, Medline (Ovid), and Scopus were used. Reference lists of identified studies were also hand searched to identify further studies, given the circumscribed review focus. Studies before the pandemic were excluded from the review. Only those published after 30 January 2020, the date when COVID-19 was declared a Public Health Emergency of International Concern by WHO in 2020, were considered. The search results and corresponding information (i.e., hits, titles, authors, and abstracts) were recorded on Microsoft Excel. Online reference management software Refworks was used for record management, and identified duplicates were removed. The SPICE Framework was used to operationalize the search strategy by guiding the review-question formulation and search process through the components setting (i.e., the context); population (i.e., target sample or perspective); interest (i.e., phenomenon of interest); comparison (i.e., comparative group—not applicable for current review); and evaluation (i.e., the result) (see Table 1). The identified databases and information sources were all searched by using the same key terms and words obtained from the literature and from the initial scoping search, with the use of Boolean Operators, as outlined in Table 2.

### 2.3. Eligibility Criteria

The studies included in this review were published from 30 January 2020 to 3 January 2022 and in English. There were no geographical restrictions, as part of inclusion. All reports screened and sought for retrieval (see Figure 1) via the identified databases were accessible through open access or through university journal subscriptions. Whilst this did not occur in our review, eligible studies that were not freely or openly accessible would have been requested or retrieved through other means to ensure an extensive systematic process. The gray literature was excluded from our review. Given the review focus area and the wealth of works in the literature and commentaries during the COVID-19 pandemic, it was deemed important to include only studies that were of high quality and were scientifically robust in order to draw reliable and accurate conclusions about N&CH staff’s experiences; as such, only peer-reviewed original studies were included. Furthermore, this decision was also made due to feasibility in consideration of the rapidity in which our review needed to be generated. Table 3 outlines additional inclusion and exclusion criteria.

### 2.4. Study Selection, Quality Assessment, and Extraction Process

After applying the relevant filters, all results retrieved were initially screened by reviewing their titles and abstracts and were documented by using a search record database recording the number of hits returned. Duplicate studies were removed, and studies which did not meet the inclusion criteria were excluded. The full text of the remaining studies was reviewed by using an inclusion tool that was developed by the reviewer. The application of the inclusion tool aided decision-making and ensured a standardized selection process.

The studies remaining were quality appraised. The Johanna Briggs Institute (JBI) Critical Appraisal Checklist for Qualitative Research [36] was used to assess quality of the qualitative studies included, a tool demonstrated to be more sensitive to aspects of validity [37]. Quantitative studies were appraised by using JBI Critical Appraisal Checklist for Analytical Cross-Sectional Studies [38] or the JBI Checklist for Prevalence Studies [39]. The use of JBI tools for quantitative prevalence and cross-sectional studies has been identified as a suitable quality appraisal tool for these types of study design and are widely used [40]. Studies with mixed-methods designs were quality assessed by using the Mixed Methods Appraisal Tool (MMAT) [41]. The MMAT has demonstrated efficacy and reliability in appraising different research designs and has been subject to revision to improve reliability [41,42].

The remaining studies (*n* = 8) were included in the review. The selected studies were read multiple times for immersion in the data of the qualitative findings [31]. A data-extraction tool was compiled for the purpose of the current review question and was used to extract relevant data and to facilitate the thematic analysis of qualitative findings.

### 2.5. Data Analysis

This mixed-methods systematic review adopted a convergent segregated approach for the analysis of included studies which initially involved the synthesis of qualitative data and quantitative data independently [43]. Qualitative data were synthesized through thematic analysis [44]. During the thematic analysis process, both first- and second-order open coding were conducted and then categorized prior to creating and defining superordinate themes and subthemes [44,45]. Quantitative data were narratively synthesized, as a meta-analysis was not possible and was beyond the scope of this review. Once independent synthesis had occurred, quantitative and qualitative data were integrated through a configuration analysis where meta-aggregation (using a thematic analysis approach) involved a simultaneous comparison of the quantitative and qualitative findings which generated themes [43]. A data transformation was not conducted.

## 3. Results

### 3.1. Number of Studies Screened and Included

From five databases (PsychINFO and PsychArticles combined) and citation searching, 472 titles and abstracts of studies were screened. Of those, the full text of 30 articles were read and assessed for eligibility. Eight studies were included in the final review; see Figure 1 for the PRISMA search flow diagram [35].

### 3.2. Characteristics of Studies and Quality Appraisal

Of the eight studies included, two studies used quantitative cross-sectional designs [28,46], four studies employed qualitative designs using a general qualitative inquiry, phenomenological, or explorative approach [47,48,49,50], and two studies were of a mixed-methods design employing a cross-sectional survey design and general qualitative inquiry approach for quantitative and qualitative aspects, respectively [51,52].

With regard to the presence of the terms MI and MD in studies, only one study explicitly used the term MI, whilst three other studies used MD or the term “moral dilemma” explicitly [28,46,47,49]. The remaining studies implied the concepts of MI and MD in their research focus and findings [48,50,51,52]. See Table 4 for study characteristics and main findings.

The quality of the included papers was generally observed to be high and generalizable to the target population. However, the mixed-methods study [52] was deemed to be of low quality due to lack of clarity regarding quantitative aspects of research methodology. This study was still included in the review due to the higher quality qualitative data presented. The overall risk of bias of studies was found to be relatively low (see Table 5). Quality and methodological issues are developed in the discussion section.

### 3.3. Quantitative Synthesis

#### 3.3.1. Levels of MI and MD

Only one study of those included directly measured MI levels in N&CH staff [28]. In the cross-sectional study conducted in the Republic of Ireland, Brady and colleagues administered the Moral Injury Events Scale (MIES) to N&CH staff and found the average score for MIES in a sample size of 390 to be 20, higher than MIES scores previously found in international hospital workers during the pandemic. Brady et al. [28] also found a significant difference between the level of MI (indicated by mean total MIES scores) in N&CH healthcare assistants (HCAs) and non-clinical staff (*p* = 0.027; adjusted *p* = 0.038). Consonant with MIES scores, 24.6% of N&CH reported poor/insufficient work ability on the Work Ability Score (WAS), and the mental health evaluation (as evidenced through assessment of traumatization) indicated levels of PTSD to be 45%. Additionally, 38.7% of workers’ scores on the WHO-5 indicated poor well-being, and suicidal ideation was reported by 13.8% of staff.

#### 3.3.2. MI and MD Psychosocial Impacts

In a quantitative study conducted during the first wave of COVID-19, Blanco-Donoso et al. [46] explored the psychological consequences of COVID-19 on N&CH workers and the impact of work stressors and job resources on secondary traumatic stress (STS), which they defined as quasi-PTSD symptoms from the exposure of the trauma experienced by another person (i.e., a patient). Similar to Brady et al. [28], Blanco-Donoso and colleagues [46] administered a range of self-report measures assessing STS, “emotional exhaustion”, “contact with death and suffering”, “social support at work”, and closed questions regarding fear of contagion (FOC) and “availability of resources” (e.g., PPE and staff). The researchers found that workload, social pressure, contact with death and suffering, and availability of resources all significantly correlated with STS and FOC. N&CH staff who had contact with residents who tested positive for COVID-19 reported significantly higher levels of STS and FOC compared to their N&CH colleagues lacking exposure, suggesting a greater likelihood of the former staff group experiencing MI or MD [13,19]. Given the significant group differences, hierarchical regression was conducted separately, controlling for exposure or non-exposure. Notably, workload, social pressure from work, and contact with death and suffering (Model 2) were found to significantly predict STS in staff exposed to COVID-19 and explained 35.5% of the variance. Model 3 comprised job resources and lack of staff and PPE, and supervisor support added 13.9% of the variance explained in STS. A further mixed-methods study conducted by Blanco-Donoso et al. [51] investigated the psychological consequences of COVID-19 on N&CH staff. However, in this study, the contributions of factors such as job demands, resources, etc., to job satisfaction were examined. Using the same scales and closed questions as their previous study, Blanco-Donoso and colleagues [51] found social pressure and emotional exhaustion to be significantly and negatively correlated with professional satisfaction, whilst contact with death and suffering, and social support were positively correlated with professional satisfaction. In the responses to the Workload Subscale of Secondary Traumatic Stress Scale (STSS) and reported emotional exhaustion, significant differences were found between staff who had been exposed to COVID-19 and staff who had been unexposed. Staff exposed to COVID-19 had significantly higher mean STSS Workload scores and emotional exhaustion scores than those who were unexposed. Furthermore, the mean total scores from STSS subscales “perceived social pressure from work” and “contact with death and suffering”, and the mean total score from the “emotional exhaustion” subscale of the Job Content Questionnaire (JCQ) were found to be significant predictors of mean total scores on the STSS subscale “professional satisfaction”. Model 1 (perceived social pressure from work and contact with death and suffering) and Model 2 (emotional exhaustion) explained 7.9% and 15.3% of the variance in professional satisfaction, respectively. This result indicates that factors of STS and emotional exhaustion may significantly impact on and contribute to the professional satisfaction N&CH feel. Interactions between fear of contagion and social support at work and contact with death and suffering and social support at work also significantly contributed to the model, explaining 6.6% of variance. Blanco-Donoso et al. [51] published calculated effect sizes for the regression analyses in contrast to their previous study. The effect for size for all significant models was relatively small. The workload and social pressure from work subscales and other items used by Blanco-Donoso et al. [51] had low Cronbach’s alpha indices, thus indicating low internal consistency and reliability (α = 0.6 and α = 0.59, respectively).

An Australian mixed-methods study presented somewhat contrasting findings to the quantitative studies [28,51]. Employing a mixed-methods survey design, Krzyzaniak et al. (2021) [52] aimed to understand how COVID-19 impacted N&CH staff. They found that over half of the respondents felt well prepared for the pandemic, and this included the feeling that there was adequate PPE and testing, information, and clear instruction. However, they found that 63% reported suffering from work-related stress.

### 3.4. Qualitative Synthesis

A thematic analysis of the included qualitative studies and qualitative aspects of mixed methods was conducted and generated the superordinate themes of resource deficits, role challenges, communication and leadership, and emotional and psychosocial consequences. Table 6 indicates the generated subthemes and the occurrence of each subtheme in the selected studies. For the purpose of this review, highlighted original quotes from the population across the studies are presented where relevant to retain and preserve the voices of the N&CH workers who participated in the studies [31].

#### 3.4.1. Resource Deficits

Resources deficits was a theme that appeared most frequently across all qualitative and mixed-methods studies, in which N&CH workers disclosed the impact on themselves and their work [47,48,49,50,51]. Through open coding and categorization of the data, subthemes of material resource deficits, time-related issues, inadequate staffing levels, and deficits in experience and knowledge were elicited [45]. In the literature, resource deficits of any nature have been perceived as a violation of what authoritative powers ought to or should be providing to ensure patient and staff safety and well-being [13,24]. It has been discussed that events or context such as resource deficits are morally injurious in that they may result leadership neglect and putting staff and patients at risk of harm to themselves and others in absence of leadership or authoritative roles “doing the right thing” and shirking responsibilities and/or accountability [13,27].

**Material Resource Deficits.** The qualitative data from all six studies indicated an inadequate or lack of material resources available to N&CH staff. Notably, N&CH staff reported limited or absent supply of personal protective equipment (PPE) (i.e., face masks or aprons) increasing fear of infection of self or other, fear of the unethical and dangerous consequences, and feelings of helplessness. One N&CH worker expressed the following:


*“Although I feel satisfied with my profession, I also feel helpless for not being able to carry out my duties effectively due to lack of resources. The measures are applied too late, and people die”*
[51] (p. 10, col. 4, lines 33–35).

That MI or MD is experienced by the staff member is implied, as they express their inability to carry out the appropriate action (i.e., N&CH duties) due to barriers from external sources (i.e., late application of measures), resulting in gross suffering and fatality, leaving them feeling helpless [10]. Staff also reported experiencing related and/or emotional repercussions caused by insufficient or delayed COVID-19 testing.

**Time-Related Issues.** The studies conducted by Blanco-Donoso et al. [51], Kryzaniak et al. [52], and Sarabia-Cobo et al. [49] reported that N&CH workers felt there was insufficient time to complete the added demands of COVID-19. N&CH workers also felt that the lack of time affected the quality of care given to residents. Due to a sense of personal responsibility, they often worked long days, significantly beyond their contracted hours:


*“The whole thing has been exhausting and extremely stressful. We had an exposure in the nursing home and that was the worst two weeks of my career. No-one contracted COVID-19 but I was working over 12 h a day for the two weeks to ensure everyone was ok”*
[52] (p. 53, col. 4, lines 22–27).

**Inadequate Staffing Levels.** N&CH staff across all the included studies expressed inadequate staffing levels to meet usual and COVID-19 demands, which they felt added pressure and had implications for their job role and psychosocial well-being. An N&CH nurse working in Belgium stated the following:


*“We had a lot of staff who were sick. We worked understaffed. We worked for two months under-staffed without adequate replacements. It was really painful and very tiring. We were very tired”*
[47] (p. 11, col. 1, lines 26–28).

**Experience and Knowledge Deficits.** It was highlighted in four out of the six studies that staff have felt unprepared for the challenges of the pandemic due to their perceived or actual inexperience and/or unfulfilled training needs [48,49,50,51]. A registered nurse working in a care home in China’s Hunan Province shared the following:


*“At the beginning of the COVID-19 pandemic, I felt panic. First, I didn’t know much about the COVID-19 virus. Then, I lacked some knowledge about epidemic prevention and control”*
[50] (p. 890, col. 2, lines 33–35).

#### 3.4.2. Role Challenges

The challenges experienced by N&CH staff in relation to their roles, such as added demands or changing responsibilities due to COVID-19, were present in some form in all studies [47,48,49,50,51,52]. An analysis generated the subthemes of change in role and demands, and conflicting principles and decision-making.

**Change in Role and Demands.** N&CH workers discussed the reorganization of their roles and increase in responsibilities due to the COVID-19 pandemic which caused them to undertake rigorous Infection Prevention Control (IPC) measures they had previously not been accustomed to [3]. N&CH workers expressed their exhaustion and stress over their newfound remits in the context of the virus, resourcing issues, and staffing issues:


*“We worked alternating shifts and were on duty here for 24, 48, or even 72 h. The workload increased a lot. We needed to disinfect the ward area, measure the body temperature of the residents three times per day, and observe whether residents had respiratory symptoms. But the number of nurses didn’t increase”*
[50] (pp. 890–891, col. 2, lines 42–43 [next page]).

**Conflicting Principles and Decision-Making**. As a result of COVID-19-induced role changes, in addition to the general pressures of geriatric care, some N&CH workers voiced that there had been instances where they were forced to make decisions that they did not understand, did not agree with, and/or that they had to make due to lack of guidance in certain situations [47,48,49,50,52]. Related to this, N&CH staff expressed the conflicting of values they encountered in their role and decision-making related to COVID-19, and which were sometimes of an unethical nature [26].


*“We were really torn between their happiness, their protection, and at the same time having to be a bit stricter and forcing them to stay in their rooms. It’s a bit abusive to tell someone we’re going to have to lock you up if you leave your room, for your safety, it doesn’t work for us, and in some cases we’ve been forced to lock them up if they’re positive, well we had to lock them up to prevent them from leaving, but it goes against the way we treat our residents”*
[47] (p. 11, col. 1, lines 17–21).

*“I cry…alone, but I cry, when another resident has died or when I watch the news, here in Catalonia there have been horrible moments, when we are told that the elderly should not go to the ICU or to the hospital…that, making those decisions, the ethical dilemma…the feeling of helplessness…that leaves anyone upset, I think we’ll all go into depression when this passes…we’ve lived through some pretty scary, scary things…I’ll never be able to forget it (cries)”* [49] (pp. 874–875, col. 2, lines 38–4 [next page]).

#### 3.4.3. Communication and Leadership

The literature on MI and MD has identified that responses and actions or absence of them by leaders is a powerful component in the development of MI or MD in employees [10,11]. All the review studies, except for the study conducted by Zhao et al. [50], documented staff experiencing feelings of non-support or feeling let down by those in charge, either at an organizational level or governmental level [47,49,51,52]. The subthemes feeling undervalued, abandoned, or betrayed; inadequate guidance and incoherent information; and concerns before COVID-19 were all generated.

**Feeling Undervalued, Abandoned, or Betrayed.** Most N&CH staff included in the studies expressed feelings that they were profoundly undervalued and unappreciated for the demanding work they performed [48,49,51,52]. N&CH staff members also spoke about feeling neglected, especially by the authoritative bodies who were meant to support them. This is well illustrated by a Spanish N&CH worker: 


*“In this work we are totally abandoned, without protective equipment or anyone who controls what is going on”*
[51] (p. 10, col. 4, lines 7–8).

**Inadequate Guidance and Incoherent Information.** Another frequent occurrence in the data was the feelings of inadequate, delayed, inconsistent, or incoherent information received from new health and social care legislation. Managers and staff felt as though the regulations were ever-changing, which made it difficult to follow and implement. One respondent shared the following: 


*“Lack of clear instructions from government/Health department”*
[52] (p. 54, col. 4, lines 3–4).

**Concerns Before COVID-19.** Some N&CH staff (including managers) stated that the present situation and challenges they faced were subsequent to years of funding cuts, low wages, infrastructural issues, and political neglect predating the COVID-19 pandemic. An N&CH worker aptly shared the following: 


*“COVID-19 has been the trigger that has exposed the deficiencies of the system”*
[51] (p. 10, col. 4, lines 26–27).

#### 3.4.4. Emotional and Psychosocial Consequences

Another significant theme—and understandable considering the previously discussed themes—was the emotional and psychosocial consequences experiences by N&CH workers [47,48,49,50,51,52]. This review identified the subthemes emotional responses, fear of contagion, and grief and loss in all the review studies.

**Emotional Responses.** The included respondent quotations from all studies signaled emotional exhaustion as a result of the N&CH challenges (relating to MI and MD) initiated or exacerbated by the COVID-19 pandemic. Staff expressed feelings of guilt, anxiety, and low mood as a consequence of their job roles. A registered nurse from a Spanish N&CH stated the following:


*“We cry together… I have cried with the residents, when the family called… I have cried of rage, of impotence and above all of sorrow, of infinite sorrow…”*
[49] (p. 875, col. 1, lines 6–8)

**Fear of Contagion.** Perhaps unsurprisingly, many N&CH staff across the studies reported a fear of being infected by and infecting others with the virus. Some N&CH staff linked their fear of contagion in N&CH with resources deficits:


*“I was scared of getting infected because of inadequate self-protection. We were short of protective equipment and wore the same mask for a few days at a time. I didn’t know if the disinfection that we usually do would be effective”*
[50] (p. 891, col. 2, lines 36–39).

**Grief and Loss.** Estimates of care home deaths during the first wave of the pandemic in the UK alone were 66,112 deaths, of which 29.3% were COVID-19 related [7]. Many N&CH staff members recounted the loss of life of residents, friends, and family members they experienced which may have been preventable but was beyond their control [49]. An N&CH employee shared the following: 


*“It is heartbreaking, and I feel guilty for not being able to remain calm and do more to save the life of my patient and the peace of mind of her children and grandchildren”*
[51] (p. 10, col. 4, lines 36–38).

### 3.5. Meta-Aggregation

An independent synthesis of quantitative and qualitative data has produced complementary findings [43]. The quantitative data indicate that there is a high presentation of MI and MD (and a high prevalence of STS) in N&CH staff population which may be linked to factors such as job resource challenges and emotional exhaustion [28,46,52]. The qualitative outcomes support the quantitative findings in that the generation of the superordinate themes (resource deficits, role challenges, communication and leadership difficulties, and emotional and psychological consequences) are reflected in the quantified data. The concordant outcomes indicate the presence of MI in N&CH workers and identifies factors that may contribute to it.

## 4. Discussion

All eight original papers included in this review set out to explore the current experiences and responses of N&CH staff during the COVID-19 pandemic. This population has often been overlooked but has experienced significant adversity [23,26,53]. The current review benefitted from the inclusion of studies across cultural and geographical boundaries, and from different study designs; thus, it offers a more comprehensive insight that enables parallels and distinctions to be drawn between different experiences of N&CHs employees [54].

The selected studies included in this review indicate a notable occurrence of MI and/or MD in N&CH staff during the COVID-19 pandemic which was inferred through the N&CH staff responses to surveys and interviews [4,5,6,7,8,9,10,11,12,13,14,15,16,17,18,19,20,21,22,23,24,25,26,27,28,29,30,31,32,33,34,35,36,37,38,39,40,41,42,43,44,45,46,47,48,49,50,51,52]. Unfortunately, a reliable and accurate prevalence of MI was not estimated in the studies; however, Brady et al. [28] found that mean MIES scores reported by N&CH staff to be significantly higher than the levels other studies had found. The authors estimated the prevalence rate of PTSD levels to be as high as 45%. Likewise, two studies conducted by Blanco-Donoso et al. [46,51] found the presence of high levels of secondary traumatic stress (STS) in N&CH populations. STS has been found to be significantly associated with MI [8], and this may imply high rates of unmeasured MI or MD in the N&CH population. The studies conducted by Blanco-Donoso and colleagues also found STS levels to be significantly predicted by workload, social pressure from work, contact with death and dying, supervisor support, job resources, and emotional exhaustion in N&CH staff. The findings of Blanco-Donoso et al. [46,51] and Brady et al. [28] support the pre-pandemic study conducted by De Veer et al. [25], indicating that MI was already a phenomenon experienced by this employee population, and findings of this review may have become more prevalent and exacerbated by the pandemic with more staff exhibiting this symptomology. Using multivariate regression analyses, the authors found that factors such as number of hours worked, job-related stress, and instrumental leadership to be significant determinants of MD, and these have also been shown to have a relationship with MI [12].

The study conducted by Brady et al. [28] offers a helpful insight into estimated MI levels in care homes during the COVID-19 pandemic. However, a limitation is that the relationship between the measured mental health levels (e.g., PTSD, suicidal ideation, and well-being) and MI in N&CH staff was unexplored, and so the impact that MI had on mental health levels and the directionality of this relationship between these constructs is unknown. It was also not clear from this study whether mental health and other difficulties preceded COVID-19. This may be a focus for future research. Brady et al. [28] also highlighted that the response rate was quite low in comparison to total number of N&CHs operating in Republic of Ireland, and this may have biased the study. The studies from Blanco-Donoso and colleagues [28,51] aimed to quantify the psychosocial consequences of the COVID-19 pandemic on N&CH staff. Whilst both studies do not explicitly state MI or MD as a focus, the findings imply the existence of both concepts and offer a helpful perspective of challenges N&CH face which may contribute to or increase the likelihood of developing of MI and MD [8,13]. As previously discussed, a limitation of these studies is that, for some of the measures used, there was low internal consistency, or some measures were unvalidated, which may influence the reliability of the results.

Kryzaniak et al. [52] attempted to explore the percentage of staff experiencing emotional consequences of the COVID-19 pandemic such as work-related stress, which they found in over half the respondents. However, in forming conclusions, it is important to note that the findings of this study must be taken with caution, as this study was generally of low quality. There was a lack of clarity in the hypothesis, methodology, and statistical analyses. Particularly notable drawbacks were that the study did not make use of robust statistical methods for meaningful conclusions to be drawn or to make generalizations about this population. Moreover, the sample size varied considerably between each questionnaire that was administered.

The quantitative findings demonstrating the impact of factors such as workload and resource inadequacy on the emotional experience of N&CH workers are consistent with the qualitative findings of the studies included [4,5,6,7,8,9,10,11,12,13,14,15,16,17,18,19,20,21,22,23,24,25,26,27,28,29,30,31,32,33,34,35,36,37,38,39,40,41,42,43,44,45,46,47,48,49,50,51,52]. The qualitative data collected highlighted the major themes of resource deficits, role challenges, communication and leadership difficulties, and emotional and psychological consequences. Whilst most of the qualitative studies did not explicitly state the concepts of MI and MD, the rich data illustrate that staff very likely experienced MI and MD during the COVID-19 pandemic. This is best illustrated by the presence of the core concepts of MI and MD in the studies, such as the perceived transgression of values being evident in N&CH staff expressing they experienced a sense of questionable and unethical decision-making either made by themselves or principle figures (e.g., care home managers or government); N&CH workers experiencing fatal consequences, such as the death of residents due to insufficient resources; witnessing and experiencing systemic failures which were deemed to have been preventable; and, as a result of this, N&CH staff feeling emotionally overwrought and unsettled [10,11,49,51].

The qualitative information is valuable and central to understanding the full and unabridged story of MI and MD in an underrepresented population. All selected studies included appropriate citations. However, some methodological flaws were present which may compromise the findings to some extent. Four out of six qualitative studies and mixed-methods studies failed to provide key and clear information about the authors’ theoretical and philosophical positions [47,48,51,52]. Some researchers did not describe their influence or potential influence on the study and vice versa [47,48,49,50,51,52]. These drawbacks may threaten the quality and validity of the findings and, in turn, the conclusions drawn from them, thus warranting further research to address these flaws.

A limitation of the quantitative and mixed-methods studies was the cross-sectional design, as a cause-and-effect relationship was not able to be established. This may be a point for future research. The quantitative studies included did not always evidence effect sizes and confidence intervals, which may have supported generation of meaningful and more robust conclusions. There was also a presence of data dredging whereby no clear hypotheses were stated, meaning that the analyses may not have been driven by the data [52,54].

The studies of all designs employed non-probability recruitment methods and used either snowballing, convenience, and/or purposive sampling strategy. Whilst this made sense with regard to the target population, this also meant that, to some degree, selection bias and sampling error may have occurred which may affect the generalizability and reliability of the studies. The survey design opened up the possibility of response bias.

As this review is concerned with MI and MD, which have ethical dimensions, fittingly it is also important to note that most studies did not discuss the risk or emotional/psychological harm to the participants (N&CH staff), and despite evidence of ethical approval being sought, there was a lack of disclosure regarding what ethical strategies were used when N&CH staff were asked about sensitive topics and PMIEs.

This review provides the most current understanding of MI and MD in N&CH workers during the COVID-19 pandemic and found that the presence of both constructs in N&CH staff predate COVID-19 pandemic but have been exacerbated by the virus [3,55]. The findings of this review seem to be congruent with previous systematic reviews conducted in this area which have found ethical and moral issues to be a recurrent theme in N&CH staff population [26,56]. Previous studies have found that MI and MD also have a direct and indirect impact on the quality of care provided to residents in N&CH settings and other vulnerable patients in other settings [26,48,57]. The implications of MI and MD on N&CH staff mental health, and the health and well-being of residents are significant and highlight the emotional and physical toll the pandemic and antecedent difficulties have had on N&CH workers. This warrants further empirical understanding in this area, in addition to prompt intervention and support for N&CH staff during and following the COVID-19 pandemic [3,13,29].

To address MI and MD, it has been suggested that a strong focus should be placed on exploring and providing moral antidotes or repairs through a collective and connected effort between governmental and institutional policy, legalization, and practice [9,13]. Whilst prevention and resolution of MI and MD requires more refinement, and there is an absence of manualized treatment for such concepts, both preventative and reparative strategies at different levels have been offered as potential solutions [9,13].

As a primary step in preventative strategies for MI and MD, it has been highlighted that all working parties connected with and involved in N&CH, including N&CH staff, must be aware of the ethical and moral “climate” within which they work, as well as the rights and responsibilities of all within that structure, and the ethical and moral consequences that may arise [30]. Effective response and communication between all parties in the N&CH has been stressed as an integral part of addressing moral and ethical concerns, whereby all parties must work in conjunction with each other, as opposed to working in isolation, in order to reinstate “moral equilibrium” [9,24,30].

In regard to the prevention of moral suffering, Greason [30] particularly emphasizes the importance of authoritative powers (for example, in the N&CH network) evaluating the foundations of the “ethical climate” so that the basis of care provided shifts away from being predominantly economically driven to being based more on best patient-centered practice, and in doing so, establishing a habitual “ethical culture”. Political legislation must mirror this and focus on the review, improvement, and maintenance of best moral practice in N&CH and to respond to causes of PMIEs such as providing adequate support and funding to N&CHs, especially in times of public health emergencies [9,18,30]. Socio-politically, it has been emphasized that health and social care policy and practice should incorporate moral philosophy and ethical considerations as routine, as COVID-19 has merely unveiled a longstanding need, as reflected in this review [18]. Akram [18] suggested that MI’s presence in healthcare professionals arises from the following of “utilitarianism” policies which need to be re-evaluated.

A proposed example of preventive strategies that N&CH institutions may take is to respond to frequent MI/MD-causing practical concerns such as resource or training deficits, for example, by providing safe and adequate levels of staffing and PPE [24]. Moreover, in regard to the prevention of moral suffering, the literature repeatedly asserts the importance of staff being listened to and responded to appropriately by those in leadership roles when moral and ethical concerns arise and/or when staff may be struggling emotionally or psychologically [8,9,10,13,30,55]. N&CH workers, as part of the health and social care system, may be reluctant to speak out and seek support, especially when they have observed or experienced an event that transgresses ethical boundaries (e.g., witnessing malpractice during the COVID-19 pandemic) [9]. Responding proactively has been identified as critical, and this may consist of those in authoritative positions (e.g., N&CH managers) encouraging and alerting staff to whistleblowing and “freedom to speak up” policies and ensuring the implementation of this; making staff aware of their employee rights; “active monitoring” of staff well-being in the workplace; providing or signposting staff to informal emotional support (e.g., peer or pastoral support); and supporting staff in seeking further evidence-based therapies if needed [9,13,30,55].

In regard to effectively remedying and repairing MI and MD that has occurred, the literature has stressed the importance of reflective practice and psychological debriefing strategies after PMIEs; acknowledgment of responsibilities and taking accountability in PMIEs; normalization of emotional and behavioral responses to PMIEs; and learning from PMIEs by actively responding and enacting changes across all N&CH levels [9,13,30,55].

Future research should aim to address methodological flaws of the existing studies; use and develop validated measures for MI and MD; and further explore the prevalence, predictors, and causes of MI and MD in N&CH staff both during and outside of the pandemic. An important research consideration will also be to conduct longitudinal studies to assess the level or experience of MI and MD at different stages of the pandemic, and how this may differ from MI and MD levels after COVID-19 vaccination rollout. Due to the limited studies and ambiguity and subjectivity surrounding the concepts of MI and MD and their meanings, it may also be beneficial to primarily establish a more refined definition [12,18]. Furthermore, there is a need for research to explore and expand on how individuals within and outside of the N&CH network may respond to moral suffering to address this longstanding “structural concern” [30,47].

### Limitations of Review

Whilst a triangulation approach was adopted by the reviewer to gain a rounded understanding of the MI and MD concept definitions prior to conducting the searches, a limitation of this review is that only one study [28] measured the presence of MI, whilst all other studies did not use the terms MI or MD as a focus of their study. A certain level of interpretation of the findings was executed by the reviewer which included comparing and contrasting the content of studies to the generally accepted definitions of MI and MD. Whilst this may have biased the studies selected, it does give prominence to the paucity of research in this specific area.

Furthermore, studies deemed to be of medium-to-low quality were not excluded from our review, in addition to most studies having methodological flaws; therefore, caution should be exercised when interpreting the findings. Again, despite this being a weakness of our review, this emphasizes the need for further high-quality and methodologically stringent research.

Whilst an extensive search of the peer-reviewed literature was conducted, the exclusion of the gray literature, as elaborated on in Section 2.3, may have contributed to publication bias. As such, it would be valuable for future reviews in this area to include the gray literature to thoroughly identify and review all relevant evidence and information [32].

## 5. Conclusions

The idea that health workers’ closely held moral beliefs and ethics can sometimes conflict with what they are expected to do in their clinical role has been around for some time. Where this conflict is felt to have been exacerbated by perceived wrongdoing or betrayal by others, emotional responses such as anxiety and guilt can, in turn, be heightened, leading to what has been termed moral injury and moral distress. This is the first systematic review of empirical studies which have addressed moral injury and moral distress in nursing and care home staff during the COVID-19 pandemic. Our synthesis of the findings showed high levels of moral injury and related constructs (secondary traumatic stress). The findings indicate that the challenges arising from the COVID-19 pandemic have led to an exacerbation of moral injury and moral distress in the healthcare staff in nursing homes. The implications are that the presence of moral injury and distress warrants prompt intervention and support for nursing and care home staff. This review selected robust and high-quality studies in this nascent empirical arena, but there is a need for further research to address some methodological flaws and to explore the prevalence, predictors, and causal relationships between variables.

## Figures and Tables

**Figure 1 ijerph-19-09593-f001:**
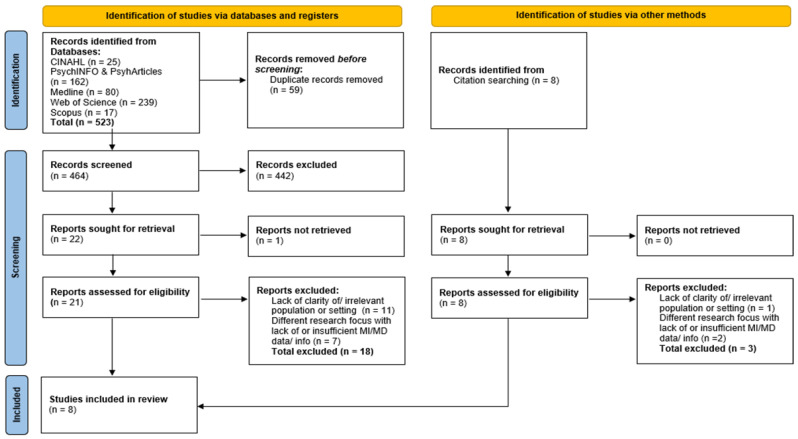
PRISMA flow diagram displaying studies that were identified, screened, and included in the review from databases, registers, and via other methods. Note: n = number of records [35].

**Table 1 ijerph-19-09593-t001:** SPICE framework and formulation.

SPICE Framework	SPICE Formulation for Review
Setting	Geriatric residential care settings (e.g., nursing and care homes)
Population	Residential care home staff (e.g., healthcare professionals and ancillary workers)
Interest	Moral injury (MI) and moral distress (MD) during COVID-19
Comparison	Not relevant to review question
Evaluation	General experience of (including prevalence, predictors, and psychological impact)

Note: SPICE framework was used for operationalization of the search strategy [34].

**Table 2 ijerph-19-09593-t002:** Database search terms using Boolean Operators.

Column Terms Combined with	Setting	Perspective/Population	Interest	Evaluation
	AND	AND	AND	AND
OR	Care home*	Staff*	Moral injur*	COVID*
OR	Nursing home*	Worker*	Morally injur*	Covid-19*
OR	Residential home*	Employee*	Moral distress*	Covid outbreak
OR	Residential care*	Member*	Morally distress*	Covid pandemic
OR	Retirement home*	Healthcare professional*	Moral* pain*	Cov-19*
OR	Convalescent home*	Healthcare support*	Moral dilemma*	2019-ncov
OR	Senior citizen* home*	Healthcare work*	Moral transgres*	Sars-cov-2*
OR	Senior* home*	Healthcare assis*	Moral* challeng*	Coronavirus*
OR	Rest home*	Healthcare support*	Moral* concern*	2020 Pandemic*
OR	Supported living*	Support work*	Moral* conflict*	
OR	Assisted living*	Nurse*	Moral agenc*	
OR	Hospice*	Care work*	Moral identit*	
OR	Palliative care*	Carer*	Moral* difficult*	
OR	Aged care*	Care assis*	Moral obligation*	
OR	Elderly*care*	Doctor*	Moral suffer*	
OR	Geriatric*	Medic*	Moral resilien*	
OR	End of life* care*	Allied health professional*	Ethical* concern*	
OR	End-of-life* care*	HCA*	Ethical dilemma*	
OR	Retirement facilit*	HCSW*	Ethical* difficult*	
OR	Residential setting*	Therapis*	Ethical transgress*	
OR	Older adult*	Manager*	Post-Traumatic Embitterment Disorder*	
OR		Admin*	PTED	
OR		Clerica*	Potentially morally injurious event*	
OR		Personal assis*	PMIE*	
OR		PA*	Moral Consequence*	
OR		Cleaner*	Moral repair*	
OR		Cleaning staff*		
OR		Non-clinical*		
OR		Ancillary*		
OR		Housekeep*		

Note: Table presenting exact review search terms with Boolean Operators which were used across all database indexes. ‘*’ after text denotes truncation of search terms to aid the search process. For example, the search term ‘Moral Injur *’ would retrieve papers containing the words moral injury, moral injuries and moral injuring.

**Table 3 ijerph-19-09593-t003:** Inclusion and exclusion criteria.

Inclusion Criteria	Exclusion Criteria
Residential care settings for older adults (e.g., N&CH or assisted-living facilities)	Non-residential care settings or residential care settings for demographics not including older people (e.g., hospital settings or care home for young people)
N&CH staff (studies which involve N&CH, as well as other populations, are included)	Other populations with no indicated participation from nursing home staff (e.g., patients, families, etc.)
MI or MD (will include studies where MI is not explicitly stated but is implicitly implied, e.g., “ethical dilemma”)	Mental health difficulties, with no indication of MI or MD (i.e., MI/MD is not implicitly implied or explicitly stated)
COVID-19 pandemic	MI and MD outside of COVID-19 or during other disease outbreaks (e.g., Ebola)
Peer-reviewed journal articles	Not peer-reviewed
Qualitative, quantitative, and mixed-methods original studies	Editorials, commentaries, symposiums, reviews, gray literature, book chapters
Published in English language	Published in languages other than English
Publication date: 30 January 2020 to 3 January 2022	Publication date before 30 January 2020

Note: N&CH = nursing and care homes, MI = moral injury, MD = moral distress.

**Table 4 ijerph-19-09593-t004:** Summary of study characteristics and main findings of included studies.

Authors and Date	Study Design and Type	Population Characteristics		Setting	Aim(s) and Methods	Main Findings
Blanco-Donoso et al. (2021a) [51]	Mixed-methods, cross-sectional survey design	Sample Size: *n* = 335. Population: N&CH staff. Mean Age: 36.12 years (SD = 10.26). Gender: 269 females, and 66 males.	N&CH Roles: 183 health and social care staff, 135 support staff, 5 center managers, and 12 unspecified roles.	Spanish N&CHs across 43 provinces	Main Aim(s): (1) To explore satisfaction levels in N&CH workers during COVID-19 amongst nursing-home workers; (2) To examine explore NHC worker’s job demands, resources, and emotional experiences. Recruitment Method: Snowball sampling. Data Collection: Secondary Traumatic Stress Scale (STSS) Workload, Social Pressure, and Professional Satisfaction Subscales; Nursing Burnout Scale (NBS); Job Content Questionnaire (JCQ) Social Support at Work Subscale Short Questionnaire of Burnout Emotion Exhaustion Subscale; Closed questions regarding lack of resources and fear of contagion developed by investigators specifically for this study; Open-ended questions. Quantitative Analyses: Pearson correlation; Independent samples t-test; Hierarchical regression, qualitative analysis, and content analysis.	Quantitative Findings: Social pressure from work and emotional exhaustion significantly and negatively related to professional satisfaction (r = −0.14, *p* < 0.05; r = −0.42, *p* < 0.01); Contact with death and suffering and social support were positively related to professional satisfaction (r = 0.20, *p* < 0.01; r = 0.25, *p* < 0.01); Significant differences in workload (t = −2.74, *p* < 0.01, d = 0.30) between N&CH staff exposed to COVID patients (M = 3.15; SD = 0.41) versus unexposed staff (M = 3.01; SD = 0.49); Significant difference emotional exhaustion between exposed and unexposed staff (t = −2.19, *p* < 0.05, d = 0.26); Social pressure from work (β = −0.18, *p* < 0.01, sr = −0.16) and contact with death and suffering (β = 0.25, *p* < 0.001, sr = 0.24) significant predictors of job satisfaction; Emotional exhaustion significantly contributed to job satisfaction (β = −0.42, *p* < 0.001, sr = −0.38). Qualitative Findings—Major Themes: (1) Inadequate working conditions and lack of resources; (2) Impact and consequences of COVID-19 crisis.
Blanco-Donoso et al. (2021b) [46]	Quantitative, cross-sectional survey design	Sample Size: N = 228. Population: N&CH workers. Mean Age: 26.29 (SD = 10.04). Gender: 183 females and45 males.	N&CH Roles: 15 doctors, 38 nurses, 44 nurse aides, 10 geriatric assistants, 19 social workers, 9 psychologists, 8 OT 5 physiotherapists, 2 management of center, and 79 unspecified roles.	N&CHs across 42 Spanish provinces	Main Aim(s): (1) To analyze psychological consequences that COVID-19 has on N&CH workers; (2) To analyze the influence that work stressors and inadequate job resources could have on the development of those consequences. Recruitment Method: Snowball sampling. Data Collection: Secondary Traumatic Stress (STS), Workload, and Social Pressure Subscales of STSS; Contact with Death and Suffering Subscale of NBS; Social Support at Work Subscale of JCQ; Closed ad hoc questions regarding lack of resources and fear of contagion (FOC) developed by investigators specifically for this study. Analyses: Pearson correlation analysis; Independent Samples *t*-tests; One-factor analysis of variance (ANOVA); Hierarchical regression.	Quantitative Findings Relevant to Review: Workload significantly correlated with STS and FOC (r = 0.40, *p* < 0.01; r = 0.19, *p* < 0.01); Social pressure significantly related to STS and FOC (r = 0.47, *p* < 0.01; r = 0.21, *p* < 0.01); Contact with death and suffering significantly correlated with STS and FOC (r = 0.45, *p* < 0.01; r = 0.27, *p* < 0.01); Lack of staff and PPE were associated with STS (r = 0.33, *p* < 0.01) and with FOC (r = 0.45, *p* < 0.01); N&CH staff in contact with COVID-19 +ve patients showed higher levels of STS than N&CH staff in N&CHs with no +ve cases detected (M = 2.80 > M = 2.62; t = 3.05, *p* < 0.01, d = 0.46); Significant mean differences in workload (F = 6.67, *p* < 0.01) and in supervisor support (F = 3.50, *p* < 0.05) found between sample group (doctors and nurses; nursing aides; other N&CH staff); STS in N&CH with COVID-19 +ve patients significantly predicted by workload (β = 0.15, *p* < 0.05), social pressure from work (β = 0.32, *p* < 0.001) and contact with death and suffering (β = 0.37, *p* < 0.001).
Brady et al. (2021) [28]	Quantitative, cross-sectional survey design	Sample Size: N = 390. Population: N&CH staff. Age: ≤30 years, 85 (21.8%); 31–50 years, 187 (47.9%); and ≥51 years, 118 (30.3%). Gender: 337 females, 50 males, and 3 other/prefer not to say.	N&CH Roles: 120 nurses, 172 HCAs, and 98 non- clinical staff.	NHI affiliated N&CHs in Republic of Ireland	Main Aim(s): (1) To quantify the mental health of N&CH staff; (2) To estimate levels of PTSD, suicidal ideation and planning, MI, coping styles, perceptions of pandemic, and work ability; (3) To explore differences of above between different types of N&CH roles. Recruitment Method: Purposive and convenience sampling. Data Collection: Impact of Events Scale (IES-R); World Health Organization Well-Being Index (WHO-5); Suicide Severity Rating Scale (C-SSRS); Moral Injury Events Scale (MIES); Coping Orientation to Problems. Experienced (Brief-COPE) Scale: 15-item questionnaire adapted from SARS study to measure health fear, social isolation, doubts about protective equipment, adequacy of training and support, and job stress; Work Ability Score (WAS), derived from the Work Ability Index (WAI). Analyses: Chi-square tests; One-way ANOVAs; Post hoc analyses-regression analyses, using GLM.	Quantitative Findings Relevant to Review: MIES mean score = 20.8 (SD = 9.1); MIES Subdomain “Transgression by others” mean = 5.9 (SD = 3.0); MIES Subdomain “Transgression by self” mean = 7.9 (SD = 4.8); MIES Subdomain “Betrayal” mean = 7.4 (SD = 4.0); There were significant differences between groups on the MIES total score (*p* = 0.027, adjusted *p* = 0.038) and the MIES “Transgression by others” subscale (*p* = 0.030, adjusted *p* = 0.048); HCAs reported a significantly higher MI level than non-clinical staff (mean difference = 3.3; SE = 1.2) and a significantly higher “Transgression by others” score than non-clinical staff (mean difference = 1.0; SE = 0.381).
Kaelen et al. (2021) [47]	Qualitative, general qualitative inquiry	Sample Size: N = 44. Population: 8 N&CH staff groups. Mean Age: Not published. Gender: 38 females and 6 males.	N&CH Roles: 10 nurses, 17 nurse aids, 9 OTs or physical therapists, and 8 non-clinical support staff.	8 Belgium N&CHs	Main Aim(s): To explore how staff perceived and experienced preparedness for addressing psychosocial and metal health needs of residents. Sampling Method: Purposive and convenience sampling. Data Collection: Focus group with N&CH staff.Analyses: Thematic content analysis.	Major Themes: (1) Incoherent information and communication; (2) Lack of personal protective equipment and testing; (3) Reorganization of work; (4) Emotional effects on staff; (5) Needs of staff.
Krzyzaniak et al. (2021) [52]	Mixed-methods survey design	Sample Size: Varies between each completed survey; N = 335–371. Population: Residential aged care facility (RACF) staff. Age Range: 20–73 years (mean age not published). Gender: Female, 320 (87%); male, 48 (13%); and other, 1 (0.3%).	N&CH Roles: 160 nurses, 16 nursing assistants, 10 other care assistants, 12 allied health professionals, 131 administrative personnel, 35 quality-and-compliance staff, 1 cleaning staff, and 1 kitchen staff.	Australian RACF	Main Aim(s): To understand the impact of the COVID-19 pandemic on the RACF workforce, including clinical, administrative, and auxiliary staff. Recruitment Method: Convenience sampling Data Collection: Adapted online survey with open-ended and closed questions about preparedness for the pandemic, information flow, PPE, management of COVID cases, visitor restriction, and other impacts on RACF staff. Quantitative Analyses Chi-square. Qualitative Analysis: Content analysis.	Quantitative: 80% (n = 290/365) of respondents felt well prepared for COVID-19; 59% felt enough PPE to look after patients appropriately (n = 219/369); 63% respondent (n = 232/368) indicated that N&CHs had adequate access to testing of residents; 92% (n = 339/368) of respondents agreed their N&CH had received sufficient information dealing +ve COVID cases; 66% (n = 243/368) indicated their N&CH had received clear instructions from official bodies about the testing of residents; 43% (n = 150/351) reported they had been unfairly or abusively treated by family or friends of residents; 52% of N&CH staff were worried about unknowingly infecting residents (n = 181/348); 63% (n = 219/349) stated they had suffered from work-related stress resulting from COVID-19; 28% (n = 97/349) indicated they were concerned about impact of pandemic on their mental health. Major Themes: (1) Personal challenges; (2) Work-related challenges.
Nyashanu et al. (2020) [48]	Qualitative, exploratory qualitative approach (EQA)	Sample Size: N = 40 (N&CH staff = 20). Population: N&CH staff and domiciliary care workers. Age Range: 25–55+ years. Gender: 20 female and 10 male.	N&CH Roles: Not published/unspecified.	Private N&CHs and domiciliary care in West Midlands, UK	Main Aim(s):To explore triggers of mental health problems among frontline healthcare professionals. Recruitment Method: Convenience sampling. Data Collection: Semi-structured interviews. Analyses: Thematic analysis and interpretive phenomenological analysis (IPA).	Major Themes: (1) Fear of infection and infecting others; (2) Lack of recognition/disparity between NHS vs. private sector conditions; (3) Lack of guidance; (4) Unsafe hospital discharges; (5) Loss of professionals and residents through deaths and staff shortages.
Sarabia-Cobo et al. (2020) [49]	Qualitative, phenomenological design	Sample Size: N = 24. Population: Registered nurses in N&CHs. Mean Age: 31.2 years (SD = 4.28). Gender: 24 female.	N&CH Roles: 24 registered nurses.	14 N&CHs across Spain, Italy, Peru, and Mexico	Main Aim(s): (1) To explore the emotional impact and experiences of registered nurses working in nursing homes; (2) To provide a perspective for designing interventions focused on emotional impact management. Recruitment Method: Purposive and snowball sampling. Data Collection: In-depth semi-structured interviews. Analyses: Inductive content analysis.	Major Themes: (1) Fear of the pandemic situation; (2) A sense of duty and commitment to care; (3) Emotional exhaustion.
Zhao et al. (2021) [50]	Qualitative, descriptive design	Sample Size: N = 21. Population: N&CH nursing staff. Mean Age: 42.7 years. Gender: 21 females.	N&CH Roles: 7 nurse managers, 7 registered nurses, and 7 nursing assistants	7 N&CHs across China’s Hunan Province	Main Aim(s): To identify challenges faced by N&CH staff during the COVID-19 pandemic. Recruitment Method: Purposive sampling. Data Collection: In-depth semi-structured interviews. Analyses: Thematic analysis.	Major themes: (1) Managing unfamiliar situations; (2) Monitoring staff; (3) Challenges arising from lack of work experience; (4) Challenges to cope with a heavy workload; (5) Challenges arising from interactions with residents and their families; (6) Challenges to control COVID-19 infection.

Note: Overview of included studies’ characteristics including components study design and type; population characteristics; setting; aims and methods; and main findings. Only study information relevant to review are displayed in the table. Irrelevant items or items which may warrant narrative exploration are excluded from table. “Unspecified role” is listed where researchers did not publish this information. “Management of center” or “management staff” refers to mainly non-clinical, administrative roles (unless otherwise stated), who may oversee and govern the day-to-day coordination and running of N&CH. Management staff may be responsible for staffing levels, supply of resources, and supporting all staff. Abbreviations: N&CH = nursing and care homes, PPE = personal protective equipment, HCA = healthcare assistant, PTSD = posttraumatic stress disorder, MI = moral injury, OT = occupational therapist, M = mean, SD = standard deviation, SE = standard error, Sr = semi-partial correlation (effect size), d = Cohen’s d for measure of effect size, +ve = positive.

**Table 5 ijerph-19-09593-t005:** Critical appraisal of included studies.

JBI Critical Appraisal Checklist for Analytical Cross-Sectional and Prevalence Studies (2020) [38,39]										
Author(s) and Date	Inclusion Criteria in Sample Clearly Defined?	Study Participants Sampled in Appropriate Way?	Sample Size and Frame Adequate?	Study Subjects and Setting Described in Detail?	Exposure Measured in Valid and Reliable Way?	Objective, Standard Criteria Used for Measurement of Condition?	Confounding Factors Identified? Strategies to Deal with Them Stated?	Outcomes Measured in a Valid and Reliable Way?	Appropriate Statistical Analysis Used?	Response rate Adequate/Managed Appropriately?
Blanco-Donoso et al. (2021b) [46]	Y	Y	U	Y	Y	Y	N	Y	Y	U
Brady et al. (2021) [28]	Y	Y	N	Y	Y	Y	N	Y	Y	U
**JBI Critical Appraisal Checklist for Qualitative Research (2020)** **[36]**										
**Author(s) and Date**	**Congruity between stated Philosophical perspective and research methodology?**	**Congruity between research methodology and research question?**	**Congruity between research methodology and data collection methods?**	**Congruity between research methodology and representation and analysis of data?**	**Congruity between research methodology and interpretation of results?**	**Statement locating researcher culturally or theoretically?**	**Influence of researcher on research and vice versa addressed?**	**Participants and their voices adequately represented?**	**Research ethical? Evidence of ethical approval?**	**Do conclusions drawn report flow from analysis/interpretation data?**
Kaelen et al. (2021) [47]	U	Y	Y	Y	Y	N	N	Y	Y	Y
Nyashanu et al. (2020) [48]	U	Y	Y	Y	Y	N	N	Y	Y	Y
Sarabia-Cobo et al. (2020) [49]	Y	Y	Y	Y	Y	N	Y	Y	Y	Y
Zhao et al. (2021) [50]	Y	Y	Y	Y	Y	N	Y	Y	Y	Y
**Mixed-Methods Appraisal Tool (MMAT) Version 2018** **[41]**										
**Author(s) and Date**	**Clear research questions?**	**Do collected data allow us to address research questions?**	**Adequate rationale for using mixed-methods design to address research question?**	**Different components of study effectively integrated to answer research question?**		**Outputs of the integration of qual and quant components adequately interpreted?**		**Divergences and inconsistencies between quant and qual results adequately addressed?**		**Different components of study adhere to quality criteria of each method tradition?**
Blanco-Donoso et al. (2021a) [51]	Y	Y	U	Y		Y		Y		Y
Krzyzaniak et al. (2021) [52]	Y	Y	U	N		U		U		N

Note: Y = yes, N = no, U = unclear.

**Table 6 ijerph-19-09593-t006:** Occurrence of superordinate themes and subthemes in selected studies.

Superordinate Themes	Subtheme (s)	Author (s) and Date					
		Blanco-Donoso et al. (2021a) [51]	Kaelen et al. (2021) [47]	Krzyzaniak et al. (2021) [52]	Nyashanu et al. (2020) [48]	Sarabia-Cobo et al. (2020) [49]	Zhao et al. (2021) [50]
Resource Deficits	Material Resource Deficits	√	√	√	√	√	√
	Time Related Issues	√	×	√	×	√	×
	Inadequate Staffing Levels	√	√	√	√	√	√
	Experience and Knowledge Deficit	√	×	×	√	√	√
Role Challenges	Change in Role and Demands	√	√	√	×	×	√
	Conflicting Principles and Decision-Making	×	√	√	√	√	√
Communication and Leadership	Feeling Undervalued, Abandoned, or Betrayed	√	×	√	√	√	×
	Inadequate Guidance and Incoherent Information	√	√	√	√	√	×
	Concerns before COVID-19	√	×	√	×	√	×
Emotional and Psychosocial Consequences	Emotional Responses	√	√	√	√	√	√
	Fear of Contagion		√	×	√	√	√
	Grief and Loss	√	×	√	√	√	×

Frequency of superordinate themes and subthemes elicited from thematic analysis (qualitative synthesis), across selected review studies with qualitative or mixed-methods design. Note: √ = indicated in study; × = not indicated in study

## Data Availability

The datasets used for this review are partly presented in the manuscript.

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
