# Peer review of "Prevalence, Predictors, and Experience of Moral Suffering in Nursing and Care Home Staff during the COVID-19 Pandemic: A Mixed-Methods Systematic Review"

_ijerph, 2022, doi:10.3390/ijerph19159593_

Round 1
Reviewer 1 Report
Overall I found this to be a thoughtful, robust and very well presented manuscript. The review methodology is comprehensive and very sound, there is a clear rationale presented for the review in the first place and the assessment of methodological quality is well conducted and articulated. I had a few points of minor clarification which I would suggest that the authors address in order to strengthen their paper. These are listed below.
Specific comments:
- When describing the review method, the authors mention that they consulted PROSPERO, however do not indicate whether their own review methodology/protocol was pre-registered here or elsewhere. Clarification on whether this was the case or why it might not have been appropriate would be useful.
- Page 3 lines 141-143 - review this for grammar - the sentence about what was included is not actually a sentence (it does not contain a main verb)
- Table 2 - were the same search terms and operators used across all databases? It's not clear from this table; although the SPICE criteria in the previous table and the list of search terms in this table were well applied.
- Eligibility criteria needs to provide a better rationale for only using things which were open access or accessible through this University's Library rather than making British Library requests or paying for access. How are the authors confident that this review is really systematic and that the studies selected are the best fit for the review aims? Especially as one of the aims is to assess prevalence. I would also like to see a stronger rationale for excluding grey literature given the subject matter of this review.
Table 4 - Blanco-Donoso et al 2021b - the participants "Physiotherapists" not "Physios" and clarify what is meant when you are discussing the Management Staff and the 6 missing values.
Discussion of resource deficits theme -while a valid theme in its own right, the links to MI and MD are not clear. The way that this is written does not fully chime with the aims of the review or seem to add any knowledge that we did not already have. I think the relevance of this for the current paper needs to be strengthened.
In the discussion section - I felt that the potential practice implications could be stronger and more specific. The discussion of how ethical and philosophical perspectives are incorporated into N&CH training was a promising start but this could have included some more detailed recommendations for practice (and potentially the ongoing education and training) of N&CH staff.
Reviewer 2 Report
Please, see the attachment.

Round 2
Reviewer 2 Report
Authors have addressed the suggested revisions very well. The paper is very clear, with useful integrations. Some minor (little) issues could however be solved, by my opinion, as follows.
Tables and figures: I suggested to include captions in all tables/figures. Authors did it. However, regarding tables, they put captions as notes. Conversely, I was meaning to add a short “title” as description at the top of them, close to the related numbers. I apologize if my suggestion was not clear in this respect.
Lines 190-192: authors write the following sentence “the full articles of 190 remaining studies were reviewed using an inclusion tool developed by the reviewer were applied to the full text articles”. This seems partly different from the one they mention in the response letter (Point 23): “the full articles of remaining studies were reviewed using an inclusion tool developed by the reviewer and the tool was applied to the full text articles”.
Line 271: a round bracket seems missing after “p = .038”.
Par 3.3.2: some lines seem repeated (286-289 and 289-292).
